REGISTERED REPORT PROTOCOL

# Validation of the moral foundations questionnaire-2 (MFQ-2) in Germany: Psychometric properties and associations with political ideology, religiosity, and personality

**Nico S. Musa**[1,2], **Sarah M. Müller**[1,2], **Frederic R. Hopp**[1,2]*

**1** Leibniz-Institute for Psychology (ZPID), Trier, Rhineland-Palatinate, Germany, **2** Trier University, Trier, Rhineland-Palatinate, Germany

* fhopp@leibniz-psychology.org

## Abstract

Cross-cultural moral psychology requires robust, validated measures. The Moral Foundations Questionnaire-2 (MFQ-2) is a recent revision of the original MFQ that offers improved assessment of moral intuitions; however, a validated German version is unavailable, limiting moral psychology research in German-speaking populations. This Stage 1 Registered Report Protocol describes the methodology for developing and psychometrically validating a German version of the MFQ-2, with a target sample size of $N \approx 1,200$. Its primary aim is to assess the instrument's factor structure using Exploratory Structural Equation Modeling/Confirmatory Factor Analysis, reliability and measurement invariance. A secondary aim is to provide initial construct validity by examining associations with psycho-social correlates, including political ideology, religiosity, personality, and ethics positions. We will test whether theoretically predicted patterns emerge in the German context. By providing a methodologically validated tool, this research will enable investigation of moral belief structures in German-speaking countries and facilitate cross-cultural comparisons in moral psychological science.

## Introduction

Moral Foundations Theory (MFT) provides a framework to capture both shared moral intuitions and their culture-specific manifestations by positing a set of universal moral domains whose relative importance varies across cultures [1]. MFT initially posits five foundational domains—Care/Harm, Fairness/Cheating, Loyalty/Betrayal, Authority/Subversion, and Purity/Degradation—each hypothesized to have evolved in response to recurrent social challenges [2,3]. Care/Harm focuses on protecting others from suffering and promoting well-being; Fairness/Cheating addresses justice and equitable treatment; Loyalty/Betrayal emphasizes

**Data availability statement:** No datasets were generated or analysed during the current study. All relevant data from this study will be made available upon study completion.

**Funding:** The author(s) received no specific funding for this work.

**Competing interests:** The authors have declared that no competing interests exist.

group allegiance and solidarity; Authority/Subversion concerns respect for social hierarchies and rules; and Purity/Degradation involves maintaining sanctity and avoiding contamination [1]. Some researchers have explored additional potential foundations, including Liberty, Honor, and Ownership, which aim to capture the broader scope of human moral reasoning, occasionally within diverse cultural contexts [4–6].

Empirical research on MFT has relied primarily on the Moral Foundations Questionnaire (MFQ). The original MFQ [7], commonly referred to as MFQ-30 or MFQ-1, facilitated extensive exploration of how political ideology and demographic factors intersect with moral priorities. Researchers revealed an asymmetry in the moral profiles of liberals and conservatives: liberals tend to strongly prioritize the individualizing foundations (Care, Fairness) over the binding foundations (Loyalty, Authority, Purity), whereas conservatives grant more equal importance to all five [2,8]. Despite its influence, the MFQ-1 exhibited suboptimal internal consistency (ranging from Cronbach's $\alpha = .24$ for Loyalty to $\alpha = .69$ for Fairness) and limited cross-cultural equivalence [9–11], especially in non-Western, Educated, Industrialized, Rich, and Democratic (WEIRD) populations [12].

The MFQ-2 was developed to address these limitations, splitting Fairness into Proportionality and Equality to improve psychometric properties [9,13]. Equality emphasizes equal treatment and should resonate more with liberal concerns for equal outcomes, whereas Proportionality focuses on fairness through merit and should align with conservative principles of meritocracy [9]. The MFQ-2 has demonstrated strong reliability and cross-cultural validity across diverse populations, including the U.S., India, Iran, and China [9,14]. This momentum is extending to Europe, with recent validations in Poland and Italy [15,16]. Beyond its psychometric strengths, the MFQ-2's new conceptual distinction is also particularly relevant for the German context, where debates around social justice and meritocracy play central roles [17]. For instance, ongoing political discussions on social welfare reforms (e.g., the 'Bürgergeld'), progressive taxation, and inheritance laws tap into the tension between ensuring equal outcomes (Equality) and rewarding proportional contributions (Proportionality). The recent electoral gains of populist parties such as the Alternative für Deutschland (AfD) further underscore the importance of instruments like the MFQ-2 for uncovering the moral foundations that drive support for such movements [18]. Despite this clear relevance and the MFQ-2's growing international adoption, a methodologically transparent and psychometrically validated German version of the MFQ-2 has not yet been established, limiting research on moral beliefs and their role in political and societal discourses in Germany, as well as the country's inclusion in cross-cultural moral psychological research.

As part of this validation, we will evaluate the MFQ-2's construct validity by examining its relationships with theoretically relevant variables. Prime candidates are political ideology and religiosity, as both represent worldviews that are deeply intertwined with moral reasoning [2]. The central role of these correlates was established in the MFQ-2's foundational development [9] and has been a focus in

subsequent validation and adaptation efforts [13,16]. To provide a more nuanced psychological characterization of the moral foundations in Germany, we extend this validation approach by also investigating associations with personality traits and ethics positions. Agreeableness, with its core of compassion and prosocial orientation, is a strong theoretical predictor for the individualizing foundations, particularly Care and Equality. Similarly, extraversion, which reflects social warmth, gregariousness, and positive emotionality, may specifically predict higher endorsement of the Care foundation [19]. Ethics positions like moral idealism and moral relativism, as operationalized in the EPQ-5, represent ethical reasoning styles that directly inform moral judgment processes and may differentially relate to individualizing versus binding foundations [20].

Based on the theoretical framework and prior validation studies, we pre-register the following hypotheses:

First, to establish the MFQ-2's structural validity and internal consistency reliability, we hypothesize:

**H1a:** The a priori six-factor structure of the MFQ-2 (Care, Equality, Proportionality, Loyalty, Authority, and Purity) will demonstrate a good fit (CFI > .95, TLI > .95, RMSEA < .06, and SRMR < .08) in the German sample, tested primarily via Exploratory Structural Equation Modeling (ESEM).

**H1b:** All six MFQ-2 subscales will demonstrate acceptable internal consistency reliability (McDonald's $\omega_t \geq .70$).

Based on prior validation studies reporting correlations ranging from r ≈ .15 to .75 [9,13], we specify a minimum effect size of r ≥ .15 for all directional hypotheses. In the following, we will test core predictions of MFT regarding the positive link between a conservative political ideology and the endorsement of the binding foundations (Proportionality, Loyalty, Authority, and Purity). This specific association is considered a defining feature of the conservative moral profile [2], leading to the following hypotheses:

**H2a–d:** A more conservative political ideology will be positively associated (r ≥ .15) with Proportionality (H2a), Loyalty (H2b), Authority (H2c), and Purity (H2d).

Beyond these linear associations, we aim to test a core pattern-difference hypothesis of MFT: that liberals do not just weigh the moral foundations differently but explicitly prioritize individualizing foundations over binding ones. We group Equality with the individualizing foundations and Proportionality with the binding foundations, following the empirically identified factor structure established in the development and validation study of the MFQ-2 [9]. To test this intra-individual priority structure, we hypothesize:

**H3:** Liberal political ideology will be significantly and positively associated with the prioritization of individualizing foundations (Care, Equality) over binding foundations (Proportionality, Loyalty, Authority, Purity), with an expected effect size of r ≥ .15.

To validate the novel distinction of Equality and Proportionality within the MFQ-2, we will test the divergent political correlates of the two Fairness foundations. While the expected positive association between a more conservative ideology and Proportionality is tested above (H2a), we hypothesize for Equality:

**H4:** A more liberal political ideology will be positively associated (r ≥ .15) with Equality.

Finally, to embed the moral foundations within the broader context of stable personality traits, we predict several associations:

**H5a–b:** Agreeableness will be positively associated (r ≥ .15) with Care (H5a) and Equality (H5b).

**H6:** Extraversion will be positively associated (r ≥ .15) with the Care foundation.

This study pursues two aims: First, we will conduct a psychometric validation of a newly developed German translation of the MFQ-2, including its factor structure, internal consistency reliability, and measurement invariance. Second, we will examine construct validity through associations with the aforementioned psycho-social characteristics. By accomplishing these aims, this study will provide a transparently developed and robustly validated instrument for assessing moral intuitions in the German language, enabling research on the moral underpinnings of German social and political life and nuanced cross-cultural comparisons.

## Methods

### Ethics information

The research protocol for this study has received ethical approval from the Ethics Committee of Trier University (EK Nr. 96/2025). Data collection complies with EU General Data Protection Regulation (GDPR). Participants' IP addresses are not stored; all data will be anonymized. Before commencing the survey, participants will be presented with an electronic consent form detailing the study's objectives, the voluntary nature of participation, data confidentiality measures, and participant rights. The survey will proceed only after explicit informed consent. Completion time is estimated at 20 minutes, and participants will be compensated at a rate of £9 per hour (approximately €10.38/h), resulting in a payment of £3 (€3,46). This is consistent with Prolific Academic's (PA) recommended hourly wage for fair compensation.

### Pilot data

No pilot data were collected for this study.

### Design

We will employ a cross-sectional survey design. The German version of the 36-item MFQ-2 will be the primary instrument [9]. We developed this version following a rigorous translation-back-translation procedure. First, two bilingual German native speakers independently translated the original English items into German. These versions were reconciled into a single preliminary version by the research team. A third bilingual speaker, who was not familiar with the original items, then translated this German version back into English. The back-translated version was compared to the original MFQ-2 to identify and resolve semantic discrepancies, ensuring conceptual and linguistic equivalence. To assess construct validity, participants will also complete several established psycho-social questionnaires.

Participants will be recruited via the online panel agency Prolific Academic (PA; https://prolific.com/). Inclusion criteria are: (1) at least 18 years of age, (2) German nationality, (3) born in Germany, (4) current residence in Germany, and (5) German as primary language, as indicated by PA's pre-screening filters. To achieve these sample characteristics, we will use PA's quota sampling feature, which also allows to approximate the sex distribution of the German population (i.e., 49% male/51% female), based on 2022 census data of the German Federal Statistical Office (Destatis; https://www.destatis.de/DE/Themen/Gesellschaft-Umwelt/Bevoelkerung/Zensus2022/_inhalt.htm). The survey will be conducted using the online survey platform Unipark. Data collection will commence following Stage 1 in-principle acceptance from PLOS One. The data collection period is scheduled to be completed within approximately six weeks from its commencement. This planned timeframe falls within the validity period of the ethical approval (Ethics Committee of Trier University, EK Nr. 96/2025).

To ensure high data quality, several measures will be implemented. At the beginning, participants must complete an AI agent detection task requiring unscrambling a five-letter nonsense word to proceed. Additionally, three instructed-response attention check items will be embedded at different points within the questionnaires. These items will appear within the first and second half of the MFQ-2, and within the EPQ-5, with clear instructions such as: *[…]; und wenn Sie dies lesen, wählen Sie jetzt bitte 'Stimme voll und ganz zu' aus.* — "[…]; and if you read this, please select 'Strongly agree'". Finally, participants will be asked to self-report AI agent use: *Haben Sie diese Umfrage mithilfe eines KI-Agenten ausgefüllt? Bitte antworten Sie ehrlich. Dies wird sich nicht auf Ihre Vergütung von Prolific Academic auswirken.* — "Did you complete this survey with the help of an AI agent? Please answer honestly. This will not affect your payment from Prolific Academic." To control for potential order effects, items of each questionnaire will be presented in randomized order for each participant, except for the political ideology and religiosity items. The questionnaires themselves will remain in a fixed sequence.

## Sampling plan

The primary objective is to gather a sufficiently large and diverse sample to robustly evaluate the factor structure of the German MFQ-2. While minimum sample size recommendations for factor analytic models often range from $N = 300$–$500$ [21,22], recent MFQ-2 validation studies suggest that larger samples are preferable. For example, the foundational development and multi-population validation study of the MFQ-2 included $N > 3,900$ participants across 19 distinct populations [9], and subsequent validations in specific cultural contexts frequently employed samples of $N \approx 800$–$1,000$ or more [13,16,23].

To formalize our sampling target, we conducted an a priori power analysis using the *semPower* and *pwr* packages in R [24,25]. For the primary internal validation, we focused on structural validity (H1a), while ensuring that the resulting sample size will also provide sufficient precision for reliability estimation (H1b). An RMSEA-based analysis indicated that a minimal sample of $N = 87$ would be sufficient to reject perfect model fit ($RMSEA_0 = 0$) in favor of a close but imperfect fit ($RMSEA_1 = .05$) with 95% power at $\alpha = .05$. A more conservative family-wise approach using a Bonferroni-adjusted $\alpha = .0033$ to account for all 15 pairwise factor correlations raised this lower bound up to $N = 382$. For the secondary correlational hypotheses (H2a–d, H3, H4, H5a–b, and H6) regarding external validity, an a priori analysis assuming a minimum effect size of $r = .15$ showed that $N = 571$ would be required to achieve 95% power.

Considering these requirements and anticipating a maximum of 15–20% exclusion rate due to attention checks and data quality screening, we target a recruitment of $N \approx 1,200$ participants to achieve a final sample of at least $N \approx 1,000$. A sensitivity analysis confirms that this planned sample will provide sufficient statistical power to detect effects as small as $r = .114$ with 95% power. This sample size thus will ensure robust evaluation of both the factor structure and the nomological network of the German MFQ-2.

## Analysis plan

**Data preparation and screening.** The raw data will be subjected to a rigorous, multi-stage screening protocol. First, participants will be excluded case-wise based on four pre-defined criteria: (1) failure of the initial AI agent detection task; (2) failure on one or more of the three embedded instructed-response attention checks; (3) invariant responding indicated by a standard deviation of less than .50 across the 36 MFQ-2 items [26]; and (4) affirmative self-report of AI agent use.

The dataset remaining after these exclusions will be screened at the variable level. We will examine the distributional properties of all MFQ-2 items by calculating their skewness and kurtosis, reporting values that exceed |2| for skewness or |7| for kurtosis. As extreme values on a Likert scale represent valid endorsements, no outliers will be removed; potential non-normality will be addressed in subsequent factor analyses using Robust Maximum Likelihood (MLR) estimation. Given the seven-point Likert scale of the MFQ-2, items will be treated as quasi-continuous, as MLR performs well with seven or more categories [27]. Since all survey items are mandatory, we anticipate no item-level missing data for completed responses. Any incidental missingness (e.g., from participant dropout or technical errors) will be handled via Full Information Maximum Likelihood (FIML) within the MLR estimator for all factor analytic models [21]. For correlational analyses, missing data will be handled via pairwise deletion to maximize data retention. After this screening, descriptive statistics will be computed for all demographic variables and the six MFQ-2 subscales to characterize the final sample. All analyses will be conducted within a Null Hypothesis Significance Testing (NHST) framework. A non-significant result ($p \geq .05$) will be interpreted as a failure to reject the null hypothesis, not as evidence for the absence of an effect.

### Structural validation: testing the factor structure of the MFQ-2

To test H1a, we will employ a split-sample cross-validation procedure. The full sample ($N \approx 1,000$) will be randomly partitioned into two subsamples ($n \approx 500$ each). In subsample 1 (i.e., calibration sample), we will assess the factor structure using Exploratory Structural Equation Modeling (ESEM), as multidimensional psychological constructs often exhibit

 

theoretically meaningful item cross-loadings due to the conceptual relatedness of the underlying factors [28,29]. This flexible approach provides a more realistic representation of the data structure and was adopted in the original MFQ-2 development [9]. Subsequently, the optimal model identified in the calibration phase will be subjected to a confirmatory replication in subsample 2 (i.e., validation sample) using Confirmatory Factor Analysis (CFA). Both ESEM and CFA will be implemented using the *lavaan* package in R [30]. For the ESEM model, a correlated factor solution will be specified using an oblique Geomin rotation.

Within the calibration sample (subsample 1), in addition to the primary six-factor model, we will also investigate alternative structural models. Based on recent findings suggesting a bidimensional structure for the Loyalty foundation, a model specifying seven first-order factors (with Loyalty split into Group Loyalty and Country Loyalty) will be tested [13]. Furthermore, consistent with MFT [1] and empirical findings [9,13], a second-order factor model will be examined using CFA. This model will test whether the first-order factors (either six or seven, depending on the best-fitting Loyalty structure) load onto two higher-order factors representing individualizing foundations (Care, Equality) and binding foundations (Proportionality, Loyalty, Authority, Purity). We will use traditional CFA for higher-order testing rather than ESEM-based extensions, which are technically more complex, prone to estimation problems, and not yet considered standard practice for higher-order validation [31].

Model fit will be assessed using established goodness-of-fit indices and cutoff criteria: the Root Mean Square Error of Approximation (RMSEA; values < .06 indicating good fit, < .08 acceptable fit), the Comparative Fit Index (CFI; values > .95 good fit, > .90 acceptable fit), the Tucker-Lewis Index (TLI; values > .95 good fit, > .90 acceptable fit), and the Standardized Root Mean square Residual (SRMR; values < .08 good fit) [21,32]. For the comparison of non-nested or differently complex models within the calibration phase (six-factor vs. seven-factor, first-order vs. second-order), information criteria such as the Akaike Information Criterion (AIC) and the Bayesian Information Criterion (BIC) will be utilized [33,34]. For either criterion, a $\Delta > 10$ will be considered to indicate strong evidence for the superior model, whereas a $\Delta < 2$ indicates that the models are practically indistinguishable [35]. In cases where the evidence is not strong ($2 \leq \Delta \leq 10$) or the criteria yield diverging conclusions, we will prioritize the BIC and the more parsimonious model (i.e., the model with fewer parameters) to avoid over-parameterization [35,36]. Fit statistics for all tested models will be reported to ensure full transparency.

Beyond global model fit, item-level diagnostics will be inspected in subsample 1. Standardized factor loadings for each item on its respective latent factor will be examined, with $\lambda < .50$ flagging potentially weak items. Additionally, items will be flagged if they demonstrate cross-loadings > .30 on any non-target factor [37] or a difference of less than .20 between the primary and any secondary loading [38]. Modification indices (MI) will be used as diagnostic tools to identify sources of local misfit. Consistent with recent validation studies [13,16], model re-specifications may be required to achieve adequate fit. Any such modification, such as the inclusion of an error covariance between items with clear semantic overlap, is contingent on a corresponding MI exceeding 20.0. A maximum of three such theoretically grounded error covariances will be considered.

To ensure cross-cultural comparability, the 36-item structure of the MFQ-2 will be maintained as the primary model and reported regardless of fit. Particular attention will be given to the Purity and Loyalty subscales, which may exhibit lower cultural salience in a secular German context. If the full 36-item model fails to achieve even marginal fit (defined as CFI/TLI < .85 or RMSEA > .10; see [21]), we will report these results transparently but may additionally explore a systematic re-specification. This would involve the removal of the single most problematic item meeting at least one of the following criteria: (a) a standardized factor loading < .40 on its primary factor, (b) the cross-loading/difference criteria described above, or (c) association with the largest remaining MI (> 10.0). Such a refinement aims to potentially identify a more stable German adaptation. All model re-specifications identified in subsample 1 will be transparently documented and then tested for their robustness in the validation sample (subsample 2).

To provide evidence for the structural validity of the final accepted model, standardized factor loadings, correlations among the latent factors, any standardized residual correlations exceeding |.10|, and the overall model fit indices will be reported and discussed.

## Internal consistency reliability

After establishing a well-fitting factor model and confirming its robustness via the split-sample procedure, we will evaluate the internal consistency reliability of each MFQ-2 subscale (and any Loyalty subfactors) using McDonald's $\omega_t$ [39] based on the full sample ($N \approx 1,000$). To test H1b, we will examine whether these coefficients reach the pre-defined threshold of $\omega_t \geq .70$, which is consistent with reliability values reported in the original MFQ-2 validation study ($\omega_t = .73$ to .95; [9]). This coefficient, derived from the standardized factor loadings, is considered a more appropriate reliability estimate than Cronbach's α for congeneric but not necessarily τ-equivalent items [40]. The use of McDonald's $\omega_t$ aligns with contemporary best practices and the reporting strategy employed in the MFQ-2 development and validation [9,13].

## Measurement invariance

While the primary study focus is the validation of the MFQ-2 within the nationwide German sample, we will also explore measurement invariance across demographic subgroups using the full sample ($N \approx 1,000$), provided the sample composition allows meaningful comparisons (e.g., across gender or different age cohorts, with each subgroup requiring a minimum size of $n = 200$ for stable model estimation). This involves testing a sequence of increasingly restrictive models: configural invariance (i.e., equivalence of factor structure), metric invariance (i.e., equivalence of factor loadings), and scalar invariance (i.e., equivalence of item intercepts) [41]. To evaluate invariance, we will primarily rely on descriptive fit index changes, as χ² difference tests are overly sensitive in large samples. Invariance will be supported if $\Delta CFI \geq -.010$ and $\Delta RMSEA \leq .015$ between successive models [42]. Standard χ² difference tests will be reported as supplementary information. Establishing scalar invariance will be necessary for making valid comparisons of latent means across groups, as underscored by findings of lacking scalar invariance for gender in a recent MFQ-2 validation study [16]. If multiple well-defined groups are available and full scalar invariance is not achieved, advanced techniques such as the alignment method [43], which was utilized for the multi-nation invariance testing in the MFQ-2 development [9], might be considered to assess approximate measurement invariance.

## Construct validity analysis

Construct validity will be assessed through pre-planned analyses using the full sample ($N \approx 1,000$) to ensure maximum statistical power for detecting the specified minimum effect size of $r \geq .15$. For all correlational analyses, we will compute Pearson's correlation coefficient ($r$). Given the large sample size, Pearson's $r$ is robust to distributional non-normality, and the Central Limit Theorem ensures appropriate sampling distributions for inference tests [44]. While Pearson's $r$ will be the standard throughout, we will descriptively report skewness and kurtosis and visually inspect Q-Q plots to identify any severe distributional anomalies.

To test the hypotheses regarding political ideology, we will examine the expected positive correlations between conservative political ideology and the binding foundations (Proportionality, Loyalty, Authority, Purity), as predicted by prior MFT research [2,8]. This involves four separate bivariate correlations between the political ideology scale and mean scores of Proportionality (H2a), Loyalty (H2b), Authority (H2c), and Purity (H2d). Furthermore, we will test the novel distinction between the two Fairness foundations by contrasting the expected conservative endorsement of Proportionality (tested in H2a) with the hypothesized positive association between a more liberal political ideology and Equality (H4).

To test H3, which predicts that liberals prioritize individualizing foundations over binding ones, we will calculate an intra-individual priority score for each participant by subtracting the composite mean of the binding foundations (Proportionality, Loyalty, Authority, Purity) from the composite mean of the individualizing foundations (Care, Equality). This priority score will be regressed on political ideology (as a continuous variable) to test whether a stronger liberal political ideology is significantly associated with a stronger prioritization of individualizing over binding foundations.

To test hypotheses related to personality, using the German version of the Big Five Inventory-2-Short Form (BFI-2-S) [45], we will investigate the anticipated positive associations between specific traits and the individualizing foundations (Care, Equality). Specifically, we will test the link between agreeableness and Care (H5a) and Equality (H5b) foundations, as well as the link between extraversion and the Care foundation (H6). All directional primary hypotheses (H2–H6) will be evaluated against the minimum threshold of $r \geq .15$.

To provide a broader assessment of construct validity, we will conduct further pre-planned analyses. First, the nomological network will be expanded by analyzing MFQ-2 associations with psycho-social variables not included in the primary hypotheses, including bivariate correlations between the six moral foundations scores and religiosity, as well as moral idealism and moral relativism, as measured by the Ethics Position Questionnaire-5 (EPQ-5) [46].

Second, to probe construct validity through group comparisons, a series of independent samples t-tests will be conducted, including comparisons based on: (1) religiosity (religious vs. non-religious participants), (2) gender, (3) geographic background (comparing participants who spent most of their lives in former East vs. West Germany or from different federal states), and (4) political orientation (comparing mean foundation scores for self-identified left-leaning vs. right-leaning participants). To maintain consistency with our primary continuous analysis (H3), this fourth comparison is intended as a supplementary descriptive analysis.

Third, to complement the bivariate analyses and address potential overlaps between predictors, we will conduct supplementary multiple regression analyses for each of the six moral foundations. In these models, all primary validation constructs (political ideology, religiosity, and personality traits) and demographic variables (gender, age) will be entered simultaneously. While bivariate correlations remain our primary benchmark for cross-cultural comparability [9,16], these multivariate models—which are consistent with recent methodological extensions in MFQ-2 validation [13]—provide a more stringent test of the MFQ-2's nomological network.

## Supporting information

**S1 Appendix. Overview of all measures and items of the study.** The file contains: (1) demographic information; (2) the full, pre-final German version of the Moral Foundations Questionnaire-2 (MFQ-2) developed for this study; (3) self-created items to assess political ideology and (4) religiosity; (5) the German versions of the Big Five Inventory-2-Short Form (BFI-2-S); (6) the German version of the Ethics Position Questionnaire-5 (EPQ-5); (7) the AI agent detection task; and (8) the three attention check items.
(DOCX)

## Author contributions

**Conceptualization:** Nico S. Musa, Frederic R. Hopp.

**Methodology:** Nico S. Musa, Sarah M. Müller, Frederic R. Hopp.

**Project administration:** Frederic R. Hopp.

**Supervision:** Frederic R. Hopp.

**Writing – original draft:** Nico S. Musa.

**Writing – review & editing:** Sarah M. Müller, Frederic R. Hopp.

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
