## [Decision Letter · Decision Letter 0]

29 Dec 2025

Dear Dr. Hopp,

Thank you for submitting your manuscript to PLOS ONE. After careful consideration, we feel that it has merit but does not fully meet PLOS ONE’s publication criteria as it currently stands. Therefore, we invite you to submit a revised version of the manuscript that addresses the points raised during the review process.

This is a valuable contribution to the literature, and the plan is generally sound and feasible.I have received one thoughtful review, and the reviewer raises a number of good points.Please ensure to address each of the reviewer's points in the revised protocol.The revised protocol may or may not be sent out for additional review after revision.

We look forward to receiving your revised manuscript.

Kind regards,

Johannes Schwabe

Academic Editor

PLOS One

**Journal Requirements:**

1. When submitting your revision, we need you to address these additional requirements. Please ensure that your manuscript meets PLOS ONE's style requirements, including those for file naming. The PLOS ONE style templates can be found at https://journals.plos.org/plosone/s/file?id=wjVg/PLOSOne_formatting_sample_main_body.pdf and https://journals.plos.org/plosone/s/file?id=ba62/PLOSOne_formatting_sample_title_authors_affiliations.pdf  2. In your cover letter, please confirm that the research you have described in your manuscript, including participant recruitment, data collection, modification, or processing, has not started and will not start until after your paper has been accepted to the journal (assuming data need to be collected or participants recruited specifically for your study). In order to proceed with your submission, you must provide confirmation. 3. In the online submission form, you indicated that your data will be submitted to a repository upon acceptance.  We strongly recommend all authors deposit their data before acceptance, as the process can be lengthy and hold up publication timelines. Please note that, though access restrictions are acceptable now, your entire minimal  dataset will need to be made freely accessible if your manuscript is accepted for publication. This policy applies to all data except where public deposition would breach compliance with the protocol approved by your research ethics board. If you are unable to adhere to our open data policy, please kindly revise your statement to explain your reasoning and we will seek the editor's input on an exemption. 4. When completing the data availability statement of the submission form, you indicated that you will make your data available on acceptance. We strongly recommend all authors decide on a data sharing plan before acceptance, as the process can be lengthy and hold up publication timelines. Please note that, though access restrictions are acceptable now, your entire data will need to be made freely accessible if your manuscript is accepted for publication. This policy applies to all data except where public deposition would breach compliance with the protocol approved by your research ethics board. If you are unable to adhere to our open data policy, please kindly revise your statement to explain your reasoning and we will seek the editor's input on an exemption. Please be assured that, once you have provided your new statement, the assessment of your exemption will not hold up the peer review process. 5. Please amend either the title on the online submission form (via Edit Submission) or the title in the manuscript so that they are identical. 6. We note that this data set consists of interview transcripts. Can you please confirm that all participants gave consent for interview transcript to be published? If they DID provide consent for these transcripts to be published, please also confirm that the transcripts do not contain any potentially identifying information (or let us know if the participants consented to having their personal details published and made publicly available). We consider the following details to be identifying information:- Names, nicknames, and initials- Age more specific than round numbers- GPS coordinates, physical addresses, IP addresses, email addresses- Information in small sample sizes (e.g. 40 students from X class in X year at X university)- Specific dates (e.g. visit dates, interview dates)- ID numbers Or, if the participants DID NOT provide consent for these transcripts to be published:- Provide a de-identified version of the data or excerpts of interview responses- Provide information regarding how these transcripts can be accessed by researchers who meet the criteria for access to confidential data, including:a) the grounds for restrictionb) the name of the ethics committee, Institutional Review Board, or third-party organization that is imposing sharing restrictions on the datac) a non-author, institutional point of contact that is able to field data access queries, in the interest of maintaining long-term data accessibility.d) Any relevant data set names, URLs, DOIs, etc. that an independent researcher would need in order to request your minimal data set. For further information on sharing data that contains sensitive participant information, please see: https://journals.plos.org/plosone/s/data-availability#loc-human-research-participant-data-and-other-sensitive-data If there are ethical, legal, or third-party restrictions upon your dataset, you must provide all of the following details (https://journals.plos.org/plosone/s/data-availability#loc-acceptable-data-access-restrictions):a) A complete description of the datasetb) The nature of the restrictions upon the data (ethical, legal, or owned by a third party) and the reasoning behind themc) The full name of the body imposing the restrictions upon your dataset (ethics committee, institution, data access committee, etc)d) If the data are owned by a third party, confirmation of whether the authors received any special privileges in accessing the data that other researchers would not havee) Direct, non-author contact information (preferably email) for the body imposing the restrictions upon the data, to which data access requests can be sent 7. If the reviewer comments include a recommendation to cite specific previously published works, please review and evaluate these publications to determine whether they are relevant and should be cited. There is no requirement to cite these works unless the editor has indicated otherwise. 

Reviewers' comments:

**Comments to the Author**

1. Does the manuscript provide a valid rationale for the proposed study, with clearly identified and justified research questions?

Reviewer #1: Yes

2. Is the protocol technically sound and planned in a manner that will lead to a meaningful outcome and allow testing the stated hypotheses?

Reviewer #1: Yes

3. Is the methodology feasible and described in sufficient detail to allow the work to be replicable?

Reviewer #1: Yes

4. Have the authors described where all data underlying the findings will be made available when the study is complete?

Reviewer #1: Yes

5. Is the manuscript presented in an intelligible fashion and written in standard English?

*PLOS ONE*

Reviewer #1: Yes

You may also provide optional suggestions and comments to authors that they might find helpful in planning their study.

**Reviewer #1:**  The manuscript presents a Stage 1 protocol for a validation study of the Moral Foundations Questionnaire-2 (MFQ-2) in Germany. The authors describe the planned study design, questionnaire, and analysis strategy. Overall, the description of the research plan seems realistic and sensible. However, I have several suggestions for improvement that might strengthen the planned research.

1) The factor structure of the MFQ-2 will be examined using exploratory and confirmatory factor analysis. It was unclear which factor analytic approach will be addressed in Hypothesis 1 (p5). It is very likely that ESEM, which allows for cross-loadings between factors, will provide a good fit to the data, whereas it might be more difficult to achieve a good fit with CFA that constrains all cross-loadings to 0. Hypothesis 1 could be more specific if, in case of ESEM, the largest size of the cross-loadings that is deemed acceptable would be specified as well.

2) I was wondering if a minimal size of the validity correlations (in addition ot their direction) could be formulated in Hypothesis 2 (p5).

3) The reliability of the MFQ-2 factors was not explicitly addressed in any hypothesis. I imagine that the measurement precision is an important criterion for evaluating an instrument. Therefore, it could be informative to specify an explicit hypothesis regarding reliability.

4) It was unclear why the authors aim for a sample size of 2,624 respondents, although their power analysis suggests that about 1,300 respondents would be enough (p9). Do they expect over 1,000 non-diligent respondents that need to be excluded? Otherwise, the rationale for this choice remained unclear. Also, I am unsure whether Prolific actually includes so many German participants that would be willing to participate. I had recently difficulty to recruit 1,000 individuals from Germany.

5) I recommend dealing with missing values using multiple imputations or full maximum likelihood. Listwise deletion is generally not recommended because this can introduce additional bias in the analyses (p10).

6) Given that the authors plan to modify the measurement model and allow for non-hypothesized residual correlations (p12), I strongly recommend splitting the sample into two random subsamples, identifying the optimal factor structure in the first subsample, and replicating it in the second subsample. This would inform whether the modifications are robust and generalize across samples. Also, it was unclear whether these analyses will be conducted using ESEM or CFA.

7) Will CFA or ESEM be used to study the higher-order factor structure (p11)? Generally, the entire manuscript remains unclear which factor specification will be followed.

8) On page 10, the authors indicate to use maximum likelihood estimation because of the ordinal response format and switch to DWLS in case of convergence problems. I believe this is incorrect. The DWLS estimator is used for ordinal responses. Generally, I do not think that an ordinal CFA is necessary for seven-point response scales. These can be treated as quasi-continuous and anaylzed with ML.

9) Model comparisons will be conducted by comparing information criteria (AIC, BIC) between models (p11). However, no information was provided on the size of the differences that will be considered meaningful. Also, what will be done if the two information criteria diverge in their conclusions?

10) If the goal of the present research is the development of a German version of the MFQ-2, then items must not be removed from the instrument (p12). Otherwise, it would be an adaptation, and the instrument would no longer be comparable to the original version.

11) It was unclear how measurement invariance across the different models (configural, metric, scalar) will be evaluated (p13). Are the authors planning to rely on inference tests, descriptive comparisons of approximate fit indices, or use some effect sizes as well?

12) The Shapiro-Wilks test will definitely be significant given the large sample (p13). I do not think that the choice of the correlation coefficient should depend on the distribution of the variable. The point estimate of the Pearson correlation does not require normally distributed data, only the inference test does.

14) Hypothesis 3 is best analyzed as a moderated regression, rather than t-test on a subsample (p13). This avoids dichotomizing a metric variable and allows analyzing the entire sample without reverting to subgroup analyses.

15) I also recommend examining the construct validity (p14) using multiple regression for all variables simultaneously as this can accommodate metric and categorical variables, without the need for an artificial dichotomization of metric variables.

**Do you want your identity to be public for this peer review?** For information about this choice, including consent withdrawal, please see our Privacy Policy

Reviewer #1: No

---

## [Author Response · Author response to Decision Letter 1]

24 Feb 2026

Drs. Johannes Schwabe and Reviewer,

Thank you for your thorough and thoughtful review of our stage 1 registered report protocol “Validation of the Moral Foundations Questionnaire-2 (MFQ-2) in Germany: Psychometric Properties and Associations with Political Ideology, Religiosity, and Personality". We have very carefully considered all of your and the reviewer’s concerns and suggestions and have revised our registered report protocol with your recommendations in mind. Below you will find detailed responses to each of your comments.

We truly believe that editor’s and reviewer’s feedback has made our registered report protocol much stronger. We genuinely thank the reviewer for pointing out the spots in which the previous iteration of this registered report protocol fell short in this regard.

To assist in readability, we have highlighted our revisions in the registered report protocol using track changes and noted the line numbers of the changes in our responses to the reviewer below. We hope that you can agree with our revisions and that our registered report protocol can be accepted in principle.

Thank you again for your time and service.

Sincerely,

Authors

Editor

Dear Dr. Hopp,

Thank you for submitting your manuscript to PLOS ONE. After careful consideration, we feel that it has merit but does not fully meet PLOS ONE’s publication criteria as it currently stands. Therefore, we invite you to submit a revised version of the manuscript that addresses the points raised during the review process.

This is a valuable contribution to the literature, and the plan is generally sound and feasible.

I have received one thoughtful review, and the reviewer raises a number of good points.

Please ensure to address each of the reviewer's points in the revised protocol.

The revised protocol may or may not be sent out for additional review after revision.

We are pleased to hear that both the editor and the reviewer see merit in our proposed study. We also thank the editor and the reviewer for highlighting points where our previous registered report protocol fell short. In addressing the reviewer’s concerns, we have clarified and revised our planned analyses in places where our previous registered report fell short.

Reviewer 1

The manuscript presents a Stage 1 protocol for a validation study of the Moral Foundations Questionnaire-2 (MFQ-2) in Germany. The authors describe the planned study design, questionnaire, and analysis strategy. Overall, the description of the research plan seems realistic and sensible. However, I have several suggestions for improvement that might strengthen the planned research.

We very much appreciate and thank the reviewer for their valuable and constructive criticism of our registered report protocol as well as for recognizing the feasibility of our proposed research design. By addressing each comment below, we hope the reviewer can agree that we have substantially improved our stage 1 registered report protocol.

R1.1. The factor structure of the MFQ-2 will be examined using exploratory and confirmatory factor analysis. It was unclear which factor analytic approach will be addressed in Hypothesis 1 (p5). It is very likely that ESEM, which allows for cross-loadings between factors, will provide a good fit to the data, whereas it might be more difficult to achieve a good fit with CFA that constrains all cross-loadings to 0. Hypothesis 1 could be more specific if, in case of ESEM, the largest size of the cross-loadings that is deemed acceptable would be specified as well.

This is an important clarification. We have revised the protocol to specify that H1a (previous H1; see R1.3.) will be primarily tested using ESEM, with CFA serving as a complementary analysis (p. 5, line 89). Regarding acceptable cross-loadings, we have now incorporated explicit thresholds into the protocol: Items will be flagged if they show cross-loadings exceeding |.30| (Hair et al., 2019) or a difference between the primary and any alternative loading of less than .20 (Howard, 2016). These criteria are outlined in the revised Structural Validation section (p. 12, lines 249–51).

R1.2. I was wondering if a minimal size of the validity correlations (in addition of their direction) could be formulated in Hypothesis 2 (p5).

We agree that specifying minimum effect sizes strengthens the hypotheses. We have revised H2a–c (and analogously, H3, H4a–b, H5a–b, and H6) to include effect size expectations. Based on prior MFQ-2 validation studies reporting correlations of r ≈ .15 to .75 (Atari et al., 2023; Zakharin & Bates, 2023), we have specified that we expect at least small-to-moderate correlations (r ≥ .15) in the hypothesized directions (pp. 5–6, lines 92–93, 97, 107, 110, 114–115; accordingly: p. 9, lines 183–186).

R1.3. The reliability of the MFQ-2 factors was not explicitly addressed in any hypothesis. I imagine that the measurement precision is an important criterion for evaluating an instrument. Therefore, it could be informative to specify an explicit hypothesis regarding reliability.

We fully agree that measurement precision is essential for evaluating an instrument. To provide a clear distinction between internal (H1a–b) and external validity (subsequent Hypotheses), we renamed the original Hypothesis 1 to H1a (p. 5, line 87; accordingly: p. 9, line 179, and p. 11, line 215) and added a new hypothesis, H1b, for reliability (p. 5 line 90–91; accordingly: p. 2, line 23, p. 5, line 85, p. 9, lines 179–180, p. 13, lines 273–275):

H1b: All six MFQ-2 subscales will demonstrate acceptable internal consistency reliability (McDonald’s ωt ≥ .70).

This minimum threshold is consistent with reliability values reported in the original MFQ-2 validation (ω = .73 to .95; Atari et al., 2023). Notably, to ensure precision, we have updated the notation from ‘ω’ to ‘ωt’ throughout the manuscript to explicitly refer to McDonald’s total omega, following the reporting standards of Atari et al. (2023).

R1.4. It was unclear why the authors aim for a sample size of 2,624 respondents, although their power analysis suggests that about 1,300 respondents would be enough (p9). Do they expect over 1,000 non-diligent respondents that need to be excluded? Otherwise, the rationale for this choice remained unclear. Also, I am unsure whether Prolific actually includes so many German participants that would be willing to participate. I had recently difficulty to recruit 1,000 individuals from Germany.

The reviewer raises a valid concern. Upon reconsideration, we agree that N = 2,624 was not well-justified. We have revised our recruitment target to N ≈ 1,200 participants. Anticipating a maximum of 15–20% exclusions due to attention checks and data quality screening, this yields a final sample of N ≈ 1,000 (p. 9, line 187–189; accordingly: p. 2, line 22). A recent pre-screening on Prolific indicates that 2,925 participants match our inclusion criteria and have been active within the past 90 days, ensuring the feasibility of our revised recruitment target.

We have updated the power analysis based on this revised target and our specified minimum effect size of r ≥ .15. These calculations confirm that a final sample of N ≥ 1,000 is more than sufficient for all planned analyses. Specifically, for the internal validation (H1a–b), N = 382 is required to detect all factor correlations with 95% power using a Bonferroni-adjusted alpha. For the correlational hypotheses (H2–H6), N = 571 is sufficient to detect our expected minimum effect of r ≥ .15 with 95% power (p. 9, lines 182–189). Furthermore, a sensitivity analysis shows that with N = 1,000, the study achieves 95% power to detect effects as small as r = .114 (p. 9, lines 189–190).

R1.5. I recommend dealing with missing values using multiple imputations or full maximum likelihood. Listwise deletion is generally not recommended because this can introduce additional bias in the analyses (p10).

The reviewer is correct that listwise deletion can be suboptimal. Hence, we have updated our strategy to maximize data retention. Given that all survey items are mandatory, missingness is only anticipated in cases of participant dropout or technical errors. To handle this robustly, we have implemented Full Information Maximum Likelihood (FIML) for all structural analyses (ESEM/CFA) and will utilize pairwise deletion for correlational analyses (p. 10, lines 207–210). This ensures that all available data points are utilized while providing more reliable estimates than traditional deletion methods.

R1.6. Given that the authors plan to modify the measurement model and allow for non-hypothesized residual correlations (p12), I strongly recommend splitting the sample into two random subsamples, identifying the optimal factor structure in the first subsample, and replicating it in the second subsample. This would inform whether the modifications are robust and generalize across samples. Also, it was unclear whether these analyses will be conducted using ESEM or CFA.

We agree that cross-validation is essential when model modifications are considered. Consequently, we have updated our analysis plan to include a split-sample procedure (p. 11, lines 215–223, p. 13, lines 264–266). The total sample (N ≈ 1,000) will be randomly divided into two subsamples (n ≈ 500 each).

Subsample 1 will serve as the calibration sample, where the factor structure will be explored and potentially refined using ESEM (addressing H1a–b). Subsample 2 will serve as the validation sample, where the final model will be tested using CFA to ensure robustness and generalizability. As our power analysis (see R1.4) indicated a minimum requirement of N = 382 for structural stability, each subsample is sufficiently powered for this procedure.

Regarding the reviewer’s second point, we clarified that ESEM will be our primary analytical framework due to its ability to account for minor cross-loadings, while CFA will be used in the validation step to test the final, more parsimonious structure (p. 11, lines 228–235). To maintain maximum power for the nomological network testing (H2–H6), these hypotheses will be tested using the full sample once structural validity has been established (p. 14, lines 297–298).

R1.7. Will CFA or ESEM be used to study the higher-order factor structure (p11)? Generally, the entire manuscript remains unclear which factor specification will be followed.

We apologize that this was unclear. We have specified the factor specifications in the revised Structural Validation section. For the first-order factor structure, ESEM will be our primary framework to account for theoretically meaningful cross-loadings, while CFA will serve as a more parsimonious complementary analysis during the validation phase (see R1.6.).

For higher-order models, we will employ traditional CFA (p. 11, lines 229, 232–234). This decision is based on the technical complexity and potential estimation problems associated with ESEM-within-CFA specifications, which are currently not considered standard practice for higher-order validation (Marsh et al., 2020).

R1.8. On page 10, the authors indicate to use maximum likelihood estimation because of the ordinal response format and switch to DWLS in case of convergence problems. I believe this is incorrect. The DWLS estimator is used for ordinal responses. Generally, I do not think that an ordinal CFA is necessary for seven-point response scales. These can be treated as quasi-continuous and anaylzed with ML.

The reviewer is correct, and we thank them for outlining this. Seven-point scales can indeed be treated as quasi-continuous (Rhemtulla et al., 2012), making MLR the most appropriate choice. We have, therefore, corrected the manuscript (p. 10, lines 204–206; deleted lines: p. 11, lines between 223–224) to specify MLR as the sole estimator and have removed references to ordinal CFA and the DWLS estimator.

R1.9. Model comparisons will be conducted by comparing information criteria (AIC, BIC) between models (p11). However, no information was provided on the size of the differences that will be considered meaningful. Also, what will be done if the two information criteria diverge in their conclusions?

We thank the reviewer for this helpful suggestion. We have added explicit criteria for model comparison to the manuscript (p. 12, lines 242–246). Following Dziak et al. (2020), we defined Δ > 10 as strong evidence for a superior model and Δ < 2 as indicating indistinguishable models. In cases of divergence between AIC and BIC or weak evidence (2 ≤ Δ ≤ 10), we will prioritize the BIC and the more parsimonious model. This strategy is specifically chosen to ensure model consistency and avoid over-parameterization in large samples (Dziak et al., 2020; Vrieze, 2012).

R1.10. If the goal of the present research is the development of a German version of the MFQ-2, then items must not be removed from the instrument (p12). Otherwise, it would be an adaptation, and the instrument would no longer be comparable to the original version.

We agree that maintaining the original 36-item structure is essential. We have revised the manuscript (pp. 12–13, lines 256–266) to state that the full 36-item model remains our primary focus and will be reported regardless of fit. Any item removal has been redefined as a last resort refinement, conducted only if marginal fit thresholds (CFI/TLI < .85, RMSEA > .10; see Kline, 2023) are not met. Therefore, the 36-item version remains our benchmark for cross-cultural comparability, while any modified configuration will be transparently labeled as a “German adaptation.”

R1.11. It was unclear how measurement invariance across the different models (configural, metric, scalar) will be evaluated (p13). Are the authors planning to rely on inference tests, descriptive comparisons of approximate fit indices, or use some effect sizes as well?

We have updated the manuscript (p. 14, lines 287–290) to clarify the evaluation of measurement invariance. To address the sensitivity of χ² difference tests in large samples (N ≈ 1,000), we will primarily rely on descriptive comparisons of fit indices. Following the recommendations of Chen (2007), invariance will be supported if the change in fit between increasingly restrictive models meets the criteria of ΔCFI ≥ -.010 and ΔRMSEA ≤ .015. Standard χ² difference tests will be reported as supplementary information. If full scalar invariance is not achieved, we will examine partial invariance or utilize the alignment method (Muthén & Asparouhov, 2014) to allow for valid latent mean comparisons.

R1.12. The Shapiro-Wilks test will definitely be significant given the large sample (p13). I do not think that the choice of the correlation coefficient should depend on the distribution of the variable. The point estimate of the Pearson correlation does not require normally distributed data, only the inference test does.

We agree with the reviewer that the Shapiro-Wilk test is likely to be oversensitive in a sample of N ≈ 1,000. Accordingly, we have removed the Shapiro-Wilk criterion and the conditional switch to Spearman’s ρ from the analysis plan (p. 14, lines 299–303). We will use Pearson’s r as the standard correlation coefficient throughout. As the reviewer notes, the point estimate of r does not require normality, and the Central Limit Theorem ensures that the sampling distribution for inference tests will be approximately normal at this sample size (Bishara & Hittner, 2012). To ensure transparency, we will report skewness and kurtosis values and use visual inspection (Q-Q plots) to identify any extreme distributional anomalies.

R1.13. Hypothesis 3 is best analyzed as a moderated regression, rather than t-test on a subsample (p13). This avoids dichotomizing a metric variable and allows analyzing the entire sample without reverting to subgroup analyses.

We agree with this methodological recommendation. We have revised our approach (p. 15, lines 310–315; accordingly: lines 331–333; see also R1.2. and p. 6, lines 105–107): H3 will now be tested using a linear regression on the full sample. Following the logic of a moderated analysis, we will use a difference score (individualizi

---

## [Decision Letter · Decision Letter 1]

9 Mar 2026

Validation of the Moral Foundations Questionnaire-2 (MFQ-2) in Germany: Psychometric properties and associations with political ideology, religiosity, and personality

PONE-D-25-62170R1

Dear Dr. Hopp,

We’re pleased to inform you that your manuscript has been judged scientifically suitable for publication and will be formally accepted for publication once it meets all outstanding technical requirements.

Kind regards,

Johannes Schwabe

Academic Editor

PLOS One

Additional Editor Comments (optional):

Your revision is comprehensive and addresses all points raised. Looking forward to the results.

Reviewers' comments:

Reviewer's Responses to Questions

**Comments to the Author**

1. Does the manuscript provide a valid rationale for the proposed study, with clearly identified and justified research questions?

Reviewer #1: Yes

2. Is the protocol technically sound and planned in a manner that will lead to a meaningful outcome and allow testing the stated hypotheses?

Reviewer #1: Yes

3. Is the methodology feasible and described in sufficient detail to allow the work to be replicable?

Reviewer #1: Yes

4. Have the authors described where all data underlying the findings will be made available when the study is complete?

Reviewer #1: Yes

5. Is the manuscript presented in an intelligible fashion and written in standard English?

*PLOS ONE*

Reviewer #1: Yes

You may also provide optional suggestions and comments to authors that they might find helpful in planning their study.

Reviewer #1: I appreciate the thorough revision by the authors. Therefore, I do not have any further comments....

**Do you want your identity to be public for this peer review?** For information about this choice, including consent withdrawal, please see our Privacy Policy

Reviewer #1: No

---

## [Editor Report · Acceptance letter]

PONE-D-25-62170R1

PLOS One

Dear Dr. Hopp,

I'm pleased to inform you that your manuscript has been deemed suitable for publication in PLOS One. Congratulations! Your manuscript is now being handed over to our production team.

Kind regards,

on behalf of

Dr. Johannes Schwabe

Academic Editor

PLOS One